# Cytosolic Quality Control of Mitochondrial Protein Precursors—The Early Stages of the Organelle Biogenesis

**DOI:** 10.3390/ijms23010007

**Published:** 2021-12-21

**Authors:** Anna M. Lenkiewicz, Magda Krakowczyk, Piotr Bragoszewski

**Affiliations:** Laboratory of Protein Homeostasis, Nencki Institute of Experimental Biology, Polish Academy of Sciences, 3 Pasteur Street, 02-093 Warsaw, Poland; a.lenkiewicz@nencki.edu.pl (A.M.L.); m.krakowczyk@nencki.edu.pl (M.K.)

**Keywords:** proteostasis, protein precursor, mitochondrial biogenesis, protein transport, protein degradation, molecular chaperone, quality control, ubiquitin, proteasome

## Abstract

With few exceptions, proteins that constitute the proteome of mitochondria originate outside of this organelle in precursor forms. Such protein precursors follow dedicated transportation paths to reach specific parts of mitochondria, where they complete their maturation and perform their functions. Mitochondrial precursor targeting and import pathways are essential to maintain proper mitochondrial function and cell survival, thus are tightly controlled at each stage. Mechanisms that sustain protein homeostasis of the cytosol play a vital role in the quality control of proteins targeted to the organelle. Starting from their synthesis, precursors are constantly chaperoned and guided to reduce the risk of premature folding, erroneous interactions, or protein damage. The ubiquitin-proteasome system provides proteolytic control that is not restricted to defective proteins but also regulates the supply of precursors to the organelle. Recent discoveries provide evidence that stress caused by the mislocalization of mitochondrial proteins may contribute to disease development. Precursors are not only subject to regulation but also modulate cytosolic machinery. Here we provide an overview of the cellular pathways that are involved in precursor maintenance and guidance at the early cytosolic stages of mitochondrial biogenesis. Moreover, we follow the circumstances in which mitochondrial protein import deregulation disturbs the cellular balance, carefully looking for rescue paths that can restore proteostasis.

## 1. Introduction

Mitochondria are cell organelles formed in the process of endosymbiosis about 1.5 billion years ago [1,2]. Since then, the former symbiote and host cell coevolved, becoming dependent on each other [2,3]. In the course of evolution, mitochondria have given up their autonomy, and most of their genetic information was transferred to the host’s nuclear genome. In turn, eukaryotic cells began to rely on the functions performed by the mitochondria, such as the efficient supply of ATP, production of multiple cofactors, or biosynthesis of lipids and amino acids. The established relationship requires continuous crosstalk between mitochondria and other cellular components [1,4]. This is possible due to mutual signaling pathways, exchange of metabolites, and integration into a cellular protein homeostasis (proteostasis) network.

Since 99% of mitochondrial proteins are encoded by the nuclear genome, the production of mitochondrial proteins depends on cytosolic ribosomes. Mitochondrial proteins are synthesized in the form of precursors that require efficient targeting, translocation, and maturation pathways to reach their intended destination and function within the organelle [5,6]. The mitochondrial protein import is essential for maintaining proper mitochondrial function and cell survival. Mitochondrial proteins are among housekeeping genes that are always expressed [7]. Consequently, deregulation of the import processes has detrimental effects on cells [8,9].

The complexity of the mitochondrial architecture, with two membranes—the outer mitochondrial membrane (OM) and the inner mitochondrial membrane (IM) that envelop two diverse aqueous compartments, namely, the intermembrane space (IMS) and the mitochondrial matrix—makes mitochondrial protein import an intricate process [6]. This complexity is multiplied by the diversity of mitochondrial proteomes that may comprise more than a thousand various proteins [10]. Still, decades of research have provided a firm understanding of how proteins are targeted, translocated, and mature within the organelle [11]. At the same time, import steps that precede precursors crossing the OM remain less investigated. In recent years, the combined efforts of many laboratories have substantially advanced our knowledge about the cytosolic fates of mitochondrial precursor proteins. These new discoveries have highlighted the direct linkage between the early stages of mitochondrial protein biogenesis and cytosolic proteostasis. Mitochondrial precursors constitute a unique class of proteins in the cytosol. They are mostly unfolded and exposed to an environment distinct from that of their destination. This renders precursor proteins among the most susceptible to damage. Thus, concerted action of multiple cytosolic protein quality control mechanisms is required to deliver precursors to mitochondria safely. Precursors are not only subject to cytosolic control but also modulate the global network of cellular proteostasis.

This review aims to provide a current overview of the rapidly evolving topic of the extramitochondrial phase of mitochondrial protein biogenesis. We concentrate on precursors’ fates from the moment of their synthesis until they enter the organelle. We describe the cellular pathways that maintain, guide, or remove precursors. We also describe how precursors affect cytosolic proteostasis and how mitochondrial import regulates cytosolic quality control machinery. Finally, we highlight the emerging links with human disease and point out open questions that await future discoveries.

## 2. One Entry but Diverse Routes of Import

Mitochondrial proteins perform diverse tasks in different parts of the organelle. Some act on their own, or more often, form part of multiprotein complexes. Some are soluble in the aqueous interior of the organelle, but many are integral to mitochondrial membranes. The proteome of human mitochondria comprises well over a thousand various proteins with often unique properties [10]. The diversity of the mitochondrial proteome is naturally duplicated by the diversity of the precursor proteins, their targeting signals, and pathways that support their import. Still, the first step of the import is shared by nearly all mitochondrial proteins. They cross the OM via the general import pore, a multi-subunit complex that serves as the receptor and a protein entry gate—the translocase of the outer membrane (TOM) [12,13,14,15]. The β-barrel protein Tom40 is the central subunit of this complex and forms the protein-conducting channel across the OM. Structural analyses revealed that the TOM complex includes two Tom40 proteins linked and organized by two Tom22 molecules and so-called small TOM subunits: Tom5, Tom6, and Tom7. TOM is not only a passage across the OM but also the regulation and precursor recognition site. Tom20 and Tom70 proteins of the TOM complex expose peripheral receptor domains that recognize internal and cleavable mitochondrial targeting signals of the precursors, and together with Tom22, help to initiate the import. Mitochondrial protein precursors are largely unfolded when passing the translocase, as the lumen of the protein-conducting channel formed by Tom40 protein would not accommodate most folded domains [12,13,16,17]. After crossing the OM via TOM, proteins are routed to their final destinations encoded within their amino acid sequence. Several protein-sorting and -assembly pathways cooperate to ensure exact targeting and maturation of the imported protein within the organelle (Figure 1).

The best-characterized and the biggest group of precursors follows the presequence pathway orchestrated by the translocase of the inner membrane, TIM23. Such precursors possess N-terminal cleavable signal sequences, also called the mitochondrial targeting signals (MTS). MTS sequences are usually 15–50 amino acid residue long, positively charged, adopt an amphipathic α-helix fold [18,19], and are primarily recognized by the Tom20 receptor. After crossing TOM, precursors are routed directly to TIM23, where the positive charge of the MTS serves to initiate electrophoretic translocation relaying on the electric potential across the IM. The membrane potential also assists the precursor’s further movement, which is finally taken over by the Hsp70-powered presequence translocase-associated motor (PAM) [20]. The recognition and import are often enhanced by internal MTS-like sequences that bind to Tom70 receptors, preserving the precursor in the import-competent state [21]. Upon import into the matrix, proteins undergo proteolytic removal of their targeting signal, an essential step in their maturation [18,22]. While most TIM23 substrates are directed into the mitochondrial matrix, the pathway also assists membrane integration of IM proteins by the lateral release of imported precursors. Some of these precursors can be subsequently released into the IMS by proteolytic cleavage [23,24]. Some presequence pathway substrates imported into the matrix can next engage the OXA translocase for their insertion into the IM [25]. A recent study uncovered yet another possible route for presequence proteins imported into the matrix. Rieske protein Rip1 requires folding by the integration of an iron–sulfur cluster, which is completed inside the matrix. Subsequently, folded Rip1 can be transferred across the IM. The transfer of the folded protein is carried by the AAA-ATPase Bcs1 [17]. Thus, Tim23 translocase forms a crossroad that supports a multidirectional import.

Another translocase of the IM is TIM22, which operates the so-called ‘carrier pathway’. Precursors of the carrier pathway contain internal targeting signals but no cleavable presequence [26]. TIM22 substrates are destined for membrane integration with multiple membrane-spanning domains. Such precursors are rich in hydrophobic amino acid residues and thus mostly recognized by Tom70 receptor. Their insertion into the IM is also assisted by the membrane potential [23,24,27].

Many proteins of the IMS utilize the mitochondrial import and assembly (MIA) pathway [28]. This pathway combines precursor import with their oxidative folding. Apart from the MIA pathway, specialized targeting routes are used by other IMS proteins.

Depending on their properties, proteins destined for the OM are built into the membrane by the sorting and assembly machinery (SAM) or by the insertase of the mitochondrial outer membrane (MIM). SAM complex integrates β-barrel proteins, which first enter via TOM and are guided by small TIM chaperones. Characteristically, substrates of MIM do not pass the TOM. Instead, these α-helical proteins are directly inserted into the IM.

For comprehensive coverage of the import pathways, refer to recent dedicated reviews [5,29,30,31]. Crucially, it is evident that mitochondrial protein translocases are not operating independently. They cooperate closely with each other, and defects in one pathway usually globally affect mitochondrial biogenesis and function [31]. On the other hand, mitochondrial fitness manifested by the maintenance of the membrane potential, optimal redox environment, or ATP levels will ultimately affect import. Thus, any disturbance within the organelle will impact the fate of the precursors in the cytosol.

Import pathways differ by their robustness, throughput, and regulation. Moreover, the efficacy of import depends on not only the route but also the specific traits of individual precursors. Thus, different molecular strategies are needed to balance the mitochondrial intake with the cytosolic supply of divers precursors.

## 3. Where Are Precursor Proteins Born?

Due to the immense disparity between cellular dimensions and sizes of individual molecules, the exact location where precursor proteins are synthesized is highly significant. Long-range transportation increases the risk of mislocalization. It also extends precursors’ exposure to the cytosolic environment, enhancing the probability of premature folding, erroneous interactions, or damage (Figure 2A). The above dangers and the need for sophisticated long-range transportation mechanisms can be greatly reduced by localized translation (Figure 2B). Multiple data evidenced that mRNAs coding mitochondrial proteins are enriched locally, suggesting that precursor synthesis can occur close to the organelle [32,33,34,35,36,37,38,39]. The need for localized translation becomes especially apparent for large cells such as neurons that span between distant parts of the body [40]. Several molecular factors that influence mRNA localization on mitochondria were identified. These include Cop1 mediated targeting [41] or the OM-associated protein Puf3, which binds 3′ untranslated region (UTR) of mRNAs in fungi [42,43] (Figure 2B). In the ovaries of *Drosophila* fly, A-kinase anchor protein Spoon/MDI was shown to recruit a translation stimulator Larp to the OM. Together, MDI and Larp amplified mRNA binding and local translation at mitochondria [44]. A recent study in human cells demonstrated that RNA-binding protein SYNJ2BP modulates levels of selected mRNAs on the OM in a stress-dependent manner regulating local translation [45].

Beyond local translation, protein synthesis can be coupled with their translocation through the membranes. First visualizations of cytosolic ribosomes revealed that they exist in two main pools, free in the solution or bound with nuclear or endoplasmic reticulum (ER) membranes. The membrane-bound ribosomes are engaged in the process of co-translational transport, in which protein synthesis is coupled with translocation into the ER. Such co-translational import proved to require early recognition of the nascent peptide provided by signal recognition particle (SRP) binding, modulated by the nascent chain associated complex (NAC). SRP binding is followed by translational stalling and precise positioning of the ribosomes at the Sec translocase complex of the ER membrane [46,47,48,49,50,51]. Only after successful docking on the translocase, the nascent protein synthesis is resumed and completed jointly with the translocation.

Cytosolic ribosomes were also observed in proximity to mitochondria, with a detailed visualization recently provided by electron cryo-tomography [52,53,54]. The early observations led to an assumption that by analogy to SRP-modulated ER translocation, the import of proteins into mitochondria is coupled with translation (Figure 2C) [11,53]. However, the following studies of mitochondrial import proved that it could occur in a fully post-translational manner. Actually, most of our mechanistic understanding of mitochondrial import pathways originates from in-organello experiments. Such assays monitor the uptake of labeled precursor proteins by isolated organelles in separation from translation [11]. Additionally, no factor corresponding to SRP, which would inseparably connect translation and translocation, was found in the case of mitochondria. Thus, co-translational import into mitochondria appears to be primarily determined by proximity, type of targeting signal, speed of polypeptide synthesis, and protein length [34,54,55,56,57]. Basically, if the N-terminal targeting signal emerges from the ribosome exit tunnel while the precursor is still being synthesized, it can engage the import machinery. Accordingly, the size of Fum1 protein, the co-translational import of which is well-documented, is above average for mitochondrial precursors [58]. Furthermore, factors such as Puf3 that bring transcripts to the mitochondrial surface can promote co-translational import [59,60,61], which could be opposed by factors maintaining mRNAs further away [62]. Moreover, ribosome binding assay experiments indicated that the OM protein OM14 can cooperate with NAC. OM14-bound NAC interacts with the ribosomal exit tunnel and the emerging polypeptide chain to regulate the co-translational import of proteins into mitochondria [63,64,65,66,67].

The co-translational import is associated with unique risks such as ribosome stalling or faulty messenger RNA. There are specialized quality control mechanisms to protect mitochondria (Figure 2D), which we describe in a separate section.

The extent of co-translational protein import into mitochondria remains uncertain and requires further studies. This is likely a regulated process that can vary depending on the cell type and its metabolic state. Most mitochondrial proteins can potentially switch between post- and co-translational import modes, but molecular determinants remain unclear. As a consequence, all types of mitochondrial precursor proteins can become exposed to the cytosol and are in need of chaperoning and quality control in this environment.

## 4. Molecular Chaperones Are Essential to Run the Post-Translational Import

During post-translational targeting, newly synthetized protein precursors are supported by molecular chaperones [68,69]. Chaperones preserve precursors in the unfolded, import-competent state, protecting them against aggregation or premature degradation. Moreover, chaperones guide precursors to the mitochondrial import complexes, helping initiate the translocation [69,70,71]. Finally, chaperones take part in the degradation of excessive or faulty precursors [72,73,74,75].

Cytosolic chaperones escort precursors during their entire journey through the cytosol. They first bind as early as during precursor synthesis and assist the precursor until it crosses the mitochondrial membrane. Although various protein partners are known to bind precursors, the general cytosolic chaperones from heat shock protein (HSP) families Hsp40, Hsp70, and Hsp90 play a central role in mitochondrial protein import (Figure 3).

Chaperones of the Hsp70 family are abundant throughout eukaryotic cells. Their chaperoning function is based on their affinity to unstructured protein segments that contain hydrophobic amino acid residues [76]. Most proteins have multiple potential Hsp70 binding sites–statistically in every 36 residues [77]. Hsp70 binding sites are readily accessible in an unfolded protein because natively internal hydrophobic amino acid residues are exposed. As mitochondrial precursor proteins need to remain unfolded before translocation, cytosolic Hsp70s play an essential role in their maintenance.

The major class of cytosolic Hsp70s is called Ssa and Hsc70/Hsp70 in fungi and other eukaryotes, respectively. It was quickly demonstrated in yeast that the depletion of Ssa1 and Ssa2 leads to the accumulation of mitochondrial precursors in the cytosol [78,79,80]. The ribosome-bound Hsp70s (called Ssb in fungi) are among the first proteins to bind newly translated polypeptide [81,82,83,84]. The majority of the mitochondria-destined nascent chains were found to interact with Ssb, while in cells lacking Ssb, some mitochondrial proteins formed aggregates [85]. Accordingly, overexpression of chaperones could overcome the growth defect caused by the excessive accumulation of precursors [86].

The binding and processivity of Hsp70 family members are driven by cycles of ATP hydrolysis. The cycles of enzymatic activity of Hsp70s are regulated by conserved co-chaperones that include J-domain proteins of the Hsp40 family and nucleotide exchange factors. The Hsp40s form a diverse group of proteins located throughout the cell [87,88]. Studies in yeast and mammals indicate that Hsp40s bind upstream of Hsp70s modulating their specificity [89,90]. In yeast, cytosolic Hsp40 chaperones such as Sis1, Xdj1, and Ydj1 were shown to recognize mitochondrial precursors and assist recruitment of Hsp70 (Ssa) chaperones [79,80,91,92,93,94].

Multiple studies demonstrated the importance of the most abundant yeast Hsp40 Ydj1 as a docking agent for mitochondrial-targeted proteins [80,91,92,93]. The *YDJ1* gene deletion affects both mitochondrial and ER targeting, causing severe growth defects [80]. Both Ydj1 and Sis1 were recently shown as essential for import of the mitochondrial β-barrel protein porin [68].

Unexpectedly, ER-located Hsp40s are also involved in routing the precursors to mitochondria along a path called ER surface-mediated protein targeting (ER-SURF). In ER-SURF, hydrophobic precursors are captured by ER-localized Hsp40s, such as Djp1, and then passed to the TOM complex pursuing their final destination [95,96]. The ER membranes provide the protective environment for fragile mitochondrial precursors. Protein transfer between ER and OM is facilitated by the tight contact of these organelles’ membranes.

While Hsp40s and Hsp70s interact with the specific motifs present in the substrate, the binding abilities of Hsp90s depend on the conformation or stability of the client protein. Hsp90s were found to contribute primarily at later stages of mitochondrial protein targeting [69,97]. Hsp90s activity is also regulated by co-chaperones that modulate such factors as ATP hydrolysis, interactions with Hsp70s, and substrate availability. The co-chaperones of Hsp90s are involved in mitochondrial precursor targeting, as exemplified by the Hsp70–Hsp90 organizing protein (HOP). HOP binds Hsp70 and Hsp90, facilitating their interaction and the transfer of precursors between these chaperones [98,99,100]. Hence, mutations of yeast HOP homolog *STI1* gene and the resulting impairment of Hsp70–Hsp90 dependent protein folding significantly lowered levels of several mitochondrial proteins [94].

Another partner for Hsp70 and Hsp90 in precursor targeting is TOMM34 protein [71]. TOMM34 was found to be equally distributed between the cytosol and the OM and involved in forming a large cytosolic complex that included HOP, Cdc37, and Hsp70/Hsp90 chaperones. Such complex was found to interact with fully translated hydrophobic mitochondrial precursors [71]. Interestingly, purified TOMM34 appeared to inhibit precursor import into the isolated yeast mitochondria [101]. Yet, a recent study elucidated the requirement for phosphorylation and cofactor-driven regulation of TOMM34 to support precursor chaperoning [72].

Cytosolic chaperone proteins dock directly to the TOM translocase linking precursors maintenance with their targeting [69,102]. Tom70 receptor of the TOM complex contains the tetratricopeptide repeat (TPR) domain that allows specific binding of Hsp70 [103]. In mammalian cells, also the Hsp90 chaperone binds this specific domain of Tom70, participating in precursor targeting [69,104]. Although the Tom20 receptor does not include the TPR domain, it also interacts with cytosolic HSPs [68,69]. Efficient chaperones docking to TOM receptors not only facilitate precursor targeting but are crucial to prevent import-related proteotoxic stress in the cytosol [105]. Moreover, TOM receptors, by interaction with the precursor, can directly perform chaperone-like stabilizing functions [21].

Apart from general chaperones, there are additional precursor-binding proteins that are involved in mitochondrial precursor targeting. Numerous mitochondrial proteins are destined to integrate into the membrane and thus contain transmembrane domains (TMD). The hydrophobic properties make the TMDs insoluble in the aquatic environment and, therefore, particularly susceptible to damage due to improper folding. Multiple sources indicate the engagement of the proteins from the ubiquilin family in the segregation and degradation of mitochondrial precursors [75,106]. Ubiquilins (UBQLN in humans), which are known to shuttle ubiquitinated proteins for proteasome degradation, have been shown to act as chaperones with a high affinity for mitochondrial precursors containing TMDs [75]. The binding of UBQLNs with precursors had a protective effect preventing TMD-driven aggregation and facilitating import into mitochondria. However, if the precursor protein was ubiquitinated, UBQLN prevented it from being imported into the mitochondria and instead promoted proteasome degradation. Deletion of mammalian ubiquilins resulted in reduced ubiquitination and degradation of the hydrophobic OM protein Omp25, and finally, accumulation of this protein’s precursors in the cytosol [75]. Furthermore, mass spectroscopy analysis in the Ubiquilin 1 knockout cells displayed cytosolic accumulation of multiple mitochondrial proteins, which was accompanied by symptoms of proteostasis stress: the general suppression of protein synthesis and cell cycle arrest [106].

Membrane-targeted mitochondrial precursors were also found to interact with TMD shielding factors of the ER [75,95,107,108]. While the environment of the ER membrane can shield mitochondrial proteins as in the ER-SURF mechanism, the interplay of protein transport machinery of the two organelles increases the risk of mistargeting. The guided entry of the tail-anchored protein (GET) complex that delivers tail-anchored proteins to ER can erroneously import mitochondria-destined TA proteins [109,110,111]. Similarly, ER proteins are sometimes erroneously integrated into the OM. Mislocalized proteins do not have their molecular partners and cannot assemble into functional complexes. Such orphan subunits that fail to complete their maturation must be extracted from the OM and targeted for degradation [109]. Interestingly, extraction and degradation processes appear to be mitochondrial and ER-associated protein quality control joint actions. The reciprocal mechanism operates at the ER, where an integral membrane protein, Ema19, promotes proteolytic degradation of not productively imported mitochondrial precursors [112].

Interdependence of the cellular protein translocation systems is a developing field, and productive targeting mechanisms might be difficult to dissect from targeting errors. Future research should disclose the depth of the integration of cellular protein logistics. In this light, reduced import efficiency of mitochondrial precursors observed in yeast cells lacking Ssh1, a paralog of the Sec61 ER translocation pore, might represent direct or indirect effects [113]. Cooperation of the import pathways is likely paralleled by the collaboration of proteolytic clearance mechanisms that prevent the accumulation of mislocalized proteins.

Ultimately, precursors integrated into the OM remain accessible for cytosolic systems, while those entering the internal environment of the organelle fall under the control of mitochondrial chaperones that act both in the IMS and matrix [5,29,30,31]. Although beyond the scope of this review, internal chaperones and assembly factors are critical to allow effective folding and maturation, enabling the transition from the precursor to the mature mitochondrial protein.

## 5. Ubiquitin-Proteasome System Degrades Mitochondrial Precursor Proteins in the Cytosol

As precursor synthesis and import are not directly coupled, the inflow of new precursors can exceed the capacity of the mitochondrial translocation machinery. Such an imbalance in precursor turnover would lead to the accumulation of unimported mitochondrial proteins in the cytosol. Mislocalized proteins jeopardize cellular functions, creating the need for an efficient and selective precursor removal mechanism to protect cytosolic protein homeostasis.

Almost half of the mitochondrial proteins were also detected in other locations in the cell. In part, these are dual localization proteins with functions that are not restricted to the mitochondria. Still, many of them might represent mislocalized precursor proteins or their quality control compartments [114,115]. From their synthesis until they cross the OM, precursor proteins are controlled by cytosolic protein clearance mechanisms. Protein degradation in the cytosol is governed by two major systems: autophagy and the ubiquitin-proteasome system (UPS). Autophagy degrades proteins in bulk, including organelles, their parts, or protein deposits. The UPS ensures the highly selective degradation of individual proteins. Substrates targeted for clearance are first tagged by the covalent attachment of small protein ubiquitin. This signal is further amplified with the formation of the polyubiquitin chains. Polyubiquitinated substrates are recognized and processed by the 26S proteasomes. Proteasomes can unfold and degrade substrate proteins, cleaving them into short peptides [116].

Attachment of ubiquitin to a substrate protein is orchestrated by the sequential action of three major types of enzymes [117,118]. First, ubiquitin is activated by the E1 ubiquitin-activating enzyme and transferred to the E2 ubiquitin-conjugating enzyme. Next, the E2 enzyme cooperates with the E3 ubiquitin ligase to transfer ubiquitin to the substrate protein. E3 proteins alone or with adaptor and scaffold proteins are responsible for substrate selection. E3s form a very large and diverse group, with 600–1000 different E3s encoded in the human genome to provide specific recognition of a broad range of cellular substrates. The ubiquitination signal can be reversed by deubiquitinating enzymes (DUBs).

Studies of the ubiquitin-conjugated proteomes found that mitochondrial proteins comprise a substantial part of the ubiquitin-conjugated cellular proteins [119,120,121,122,123]. Mitochondrial proteins, except for the OM ones, are exposed to the ubiquitination machinery primarily as precursors. This goes in line with the observation that newly synthesized proteins compose a significant part of the substrates of UPS [124]. Proteins are frequently ubiquitinated as early as during their synthesis, with mitochondrial precursors among these [125,126].

Therefore, UPS becomes an important regulator of precursor protein levels and their availability for mitochondrial import, as evidenced by multiple studies [127,128,129,130,131,132,133,134]. The ubiquitination, followed by the degradation, prevents the accumulation of mitochondrial proteins in the incorrect compartment and reduces proteostasis strain. Many proteins destined for mitochondria are degraded by the proteasome when their import is defective or slows down [127,128,129,131,133,134,135]. However, the precursor protein removal is not limited to import defects, with a significant proportion of precursor proteins being continuously degraded. Consistently, inhibition of the proteasome results in increased precursor uptake and accumulation inside the mitochondria [127,128,129,133,134]. Therefore, precursors rescued from UPS degradation may not be protein waste but functional proteins. The continuous competition of the mitochondrial import apparatus and the UPS for precursor proteins effectively prevents their cytosolic accumulation [129,133].

Interestingly, recent data identify the nucleus as an important destination for the non-imported mitochondrial precursors [136]. The nucleus is also the location for a significant portion of cellular proteasomes [137]. With the newly described pathway, termed nuclear-associated mitoprotein degradation (mitoNUC), unimported MTS-containing substrates of the TIM23 pathway are routed to the nucleus, where the proteasome processes them (Figure 4) [136]. The nuclear degradation is preceded by substrate ubiquitination by three ubiquitin ligases: San1, Ubr1, and Doa10 [136].

The mitoNUC pathway requires the presence of MTS on its substrates. However, it remains to be determined whether the UPS uses MTS sequences as a common recognition trait. Our study in yeast showed that the proteasome regulates a large number of proteins targeted to the IMS via the MIA pathway. However, although MIA substrates share conserved cysteine motifs, and other common sequence and structural features, their degradation did not depend on a single specific E3 enzyme. Instead, different and redundant ubiquitin ligases provided ubiquitination of MIA precursor proteins [130]. Thus, no common recognition mechanism seems to exist in the case of precursors that follow the MIA pathway. Further studies demonstrated that some mitochondrial precursors might have stabilizing regions that prevent their degradation and allow for slow import [138]. Precursors might also be processed in the cytosol, reducing their stability. A recent study found that dipeptidyl peptidases DPP8/9 process the N-terminal part of the adenylate kinase AK2 precursor, causing its proteasome-mediated destabilization [133]. Similar modifications might also regulate other precursors. The partially unfolded state that exposes large parts of the polypeptide sequence is likely a general destabilizing property of most precursors, especially those enriched with hydrophobic residues.

Although multiple ubiquitin ligases were found to mediate the ubiquitination of specific mitochondrial proteins, deciphering precursors’ internal signals for degradation and understanding intricate recognition mechanisms requires further research [139,140,141,142]. It should be noted that ubiquitin attachment affects precursor proteins not only by targeting them for degradation. Recent studies showed ubiquitination as a direct regulator of the import process as even a single ubiquitin moiety can directly interfere with the transport of the substrate through the OM [130]. In human cells, ubiquitinating and deubiquitinating enzymes March5 and USP30 were shown to act adjacent to the TOM translocase to determine if the precursor can continue its import [132]. In contrast to the precursor degradation, the ubiquitin attachment is a reversible process that significantly increases the versatility of precursor regulation by the UPS.

Later parts of the review describe specific quality control mechanisms, many of which are linked by the involvement of the UPS.

## 6. Disturbance and Restoration of Proteostasis in the Cytosol

Despite continuous guidance and control provided by chaperones and clearance pathways, levels of the precursor proteins can build up outside mitochondria. Such buildup occurs when the influx of new precursors significantly exceeds the combined capacity of the import machinery and regular clearance mechanisms. Such precursor turnover imbalance can have different sources. Protein import is highly regulated and can even be paused [143,144]. A stressful stimulus may reduce mitochondrial membrane potential and consequently trigger an acute import deficiency [136,145,146]. Mitochondrial import deficiency may result from mutations in the components of translocases or defective precursors that stall during the translocation. All of these events lead to the accumulation of non-imported precursors in the cytosol and, in consequence, proteotoxic stress, called mitochondrial precursor overaccumulation stress (mPOS) [86,147]. Cells are equipped with several tiers of responses to alleviate such stress and restore homeostasis [9,86,148,149]. Initial responses involve regulation of existing protein turnover machinery and are followed by the transcriptome adjustment.

The immediate responses to import failure, which do not require transcription reprogramming, include the protective mechanism called unfolded protein response activated by mistargeting of proteins (UPRam) (Figure 4) [148]. During UPRam, accumulation of unimported precursors in the cytosol promotes proteasome assembly via assembly factors Irc25 and Poc4 [148]. This results in increased proteasomal degradative capacity, which supports the clearance of mislocalized precursors from the cytosol [86,148]. In parallel, mPOS results in translation attenuation, limiting further precursor supply [86,148]. The translation attenuation and remodeling can be triggered rapidly by phosphorylation of the translation initiation factor eIF2α, a vital response to many forms of stress [150]. In the case of mPOS, the translation is further attenuated at the levels of nuclear export of ribosome components, their assembly, and the expression of ribosomal subunits [86,148]. During mPOS in yeast, Nog2 and Gis2 proteins are upregulated [86]. Nog2 inhibits the nuclear export of the 60S ribosomal subunits. The lack of 60S transport reduces the level of functional cytosolic 80S ribosomes and, finally, protein translation [151]. Simultaneously, Gis2 promotes cap-independent translation allowing the expression of selected genes [86].

Similarly to translation reduction, proteasome activation is not limited to increased assembly and is modulated on multiple levels. In response to mPOS, cells boost the production of proteasome components. This was found to be mediated by Rpn4, a key regulator of proteasome expression [152]. The upregulation of proteasome is a part of a broader transcriptional program named mitoprotein-induced stress response [8,149,152]. The mitoprotein-induced stress response depends on the major heat shock transcription factor Hsf1 and thus associates with the general cellular defense strategy against proteotoxic stress [149,152]. Hsf1 under stress conditions is released from chaperon complexes to induce transcriptional reprogramming. This results in the induction of many chaperones and Rpn4 [152]. Moreover, the mitoprotein-induced stress response reduces the expression of many mitochondrial proteins to decrease the protein load on mitochondrial translocases.

By adjusting both synthesis and degradation, stress responses are plainly aimed at balancing the precursor turnover and effectively limiting the consequences of their mislocalization. Such responses are not unique to mPOS and fall into the general proteostasis maintenance programs. The so-called integrated stress response (ISR), which downregulates general protein synthesis but upregulates specific genes in response to intrinsic or extrinsic stresses, is conserved in eukaryotic cells [153,154]. Suppressed cytosolic translation during proteotoxic stress enables the cell to concentrate on producing only selected proteins that are essential for supporting cell survival [155,156,157].

Protein misfolding inside mitochondria triggers a mechanism called mitochondrial unfolded protein response (UPRmt) [158,159]. This pathway, first characterized in Caenorhabditis elegans, reacts to proteotoxic stress in mitochondria by increasing the expression of mitochondrial chaperones and proteases in an ATFS-1-dependent manner (Figure 4). Interestingly, the activation of UPRmt is directly connected with the function of mitochondrial protein import. In physiological conditions, the precursor of ATFS-1 is imported into the mitochondrial matrix, where it is degraded by LON AAA protease. However, when the mitochondrial import fails, ATFS-1 takes another route and is imported into the nucleus. In the nucleus, ATFS-1 acts as a transcriptional factor inducing the expression of genes encoding proteins implicated in UPRmt, such as mtHsp60 or mitochondrial protease ClpP [145,158]. Moreover, ATFS-1 helps balance the expression of the genes involved in oxidative phosphorylation, reducing the load of new proteins [160]. As ATFS-1 action depends on its precursor import, UPR^mt^ is likely to be triggered in the conditions that promote precursor mislocalization. Thus, UPR^mt^ will likely accompany other mPOS-related mechanisms.

UPR^mt^ is not restricted to nematodes. In yeasts that do not contain ATFS-1 homologs, the HAP complex plays a similar role by regulating genes encoding respiratory components [4,161,162]. In human cells, regulation of gene expression in the UPR^mt^ depends on the interplay of ATF4, ATF5, and DDIT3/CHOP transcription factors and thus is integrated with the ISR [158,160,163,164]. Recently, a proteolytic release of DELE1 protein by mitochondrial protease OMA1 was shown to signal stress from mitochondria to the cytosol in mammalian cells. In the cytosol, DELE1 promotes eIF2α phosphorylation by the HRI kinase, triggering translation reprogramming [165,166,167]. Phosphorylated eIF2α prioritizes the expression of ATF4, ATF5, and CHOP, increasing the expression of chaperones and proteases involved in protein degradation [165,166]. Another study found that during mitochondrial dysfunction, CHOP, along with C/EBPβ, modulates ATF4 levels, tuning down the ATF4-dependent transcriptional program. Thus, CHOP acts as a regulator that tunes down ISR, minimizing negative effects that arise when the response is prolonged [168]. Additionally, other proteins such as sirtuin SIRT3 were shown to modulate mitochondrial stress signaling [169].

Although mitochondrial stress’s intricate regulation and signaling have not been fully resolved, cellular responses to internal mitochondrial malfunction are likely to converge with these related to mPOS. Disturbed targeting of mitochondrial proteins can be both a cause and a result of mitochondrial stress. Importantly, mitochondrial stress has a far-reaching impact, regulating lifespan and healthspan and contributing to pathology [170,171].

## 7. Mitochondrial Precursor Proteins Are Prone to Aggregation

Many mitochondrial proteins are metastable proteins that are close to their solubility limit [172,173]. Metastable proteins are prone to aggregation if their concentration increases. Hence, the aggregates that accumulate in the cell after proteasome inhibition are enriched with mitochondrial precursor proteins [174]. Moreover, if protein aggregates are present in the cell, mitochondrial precursor proteins easily co-aggregate, limiting their availability for organelle biogenesis [175]. A similar observation was made in the study of Tom1 and HUWE1 ubiquitin ligases that guard the levels of non-complex ribosomal proteins in yeast and human cells, respectively [176]. The defect in this pathway increased the formation of protein aggregates enriched in both ribosomal and mitochondrial proteins.

In certain conditions, protective mechanisms that adjust the influx of mitochondrial precursor proteins and modulate degradation systems are insufficient. If these mechanisms fail, unimported mitochondrial precursors may form aggregate-like deposits [136,177] (Figure 4). Collecting excess precursors within insoluble deposits might reduce their toxicity and allow for safer storage [136,178]. However, such deposits, initially formed by most unstable precursors, can cause further cytosolic aggregation of other proteins. The potential to form cytosolic deposits of mitochondrial precursors is significant in cells susceptible to protein aggregation, such as neurons. Significantly, mitochondrial precursor aggregates were shown to increase misfolding of α-synuclein or amyloid β proteins [177], oligomerization and aggregation of which are responsible for Parkinson’s and Alzheimer’s diseases.

In response to precursor aggregation, cells upregulate specific molecular chaperones at the transcriptomic and protein levels [177]. In parallel, the overexpression of Mia40 protein and the resulting increase in protein import capacity were shown to suppress aggregation of an aggregation-prone protein [179]. A possible explanation is that precursors compete with other unstable proteins for cytosolic chaperones, and combined, they can exceed the capacity of the cytosolic folding machinery. Thus, both upregulation of chaperones and import rate could be effective strategies to prevent aggregation.

Protein misfolding is a pivotal factor in degenerative disorders, including Alzheimer’s, Parkinson’s, or diabetes. The role of mitochondrial protein import in such conditions is still ambiguous. For example, Alzheimer’s associated variant of amyloid-β was found to co-aggregate with mitochondrial precursors and accumulated in TOM and TIM23 complexes [175,180]. On the other hand, the resulting mitochondrial stress can trigger the UPRmt, which reduces the toxicity of amyloid-β peptide in *C. elegans* Alzheimer’s disease model [181].

Precursor aggregation is a new area requiring further research to understand its placement in the proteostasis network and pathology. Possibly, different deposits are formed, some of which are beneficial as storage compartments, while others are harmful to the protein folding in the cytosol. Additionally, an open issue is how aggregated precursors could be cleared. The proteasome degrades proteins singly, and thus protein aggregates may escape proteasomal control. However, a recent study showed that UBQLN2 works with HSP70-HSP110 disaggregase, allowing proteasomal targeting of proteins that are removed from aggregates [182]. This observation, combined with the high affinity of UBQLNs towards mitochondrial precursors, creates the possibility that UBQLNs allow the import of disaggregated precursor proteins into the mitochondria. Albeit speculative, this concept can be linked to another observation in which aggregates formed by newly synthesized proteins were found attached to mitochondria [183].

## 8. Restoration of Import through the Outer Mitochondrial Membrane

Not all import events are successfully completed, even under physiological conditions. It is well-established that the presequence-containing precursors can be arrested at a late stage of their import into the matrix if their C-terminal part contains a tightly folded domain [184,185,186,187,188,189,190,191,192,193,194]. Early mitochondrial proteome analyzes detected precursor forms of matrix and IM targeted proteins in association with the cytosolic side of the OM [195], likely in part representing such failed translocation events. Precursors that stall during import can span both outer and inner membranes in a stable complex with TOM and TIM23 translocases. Most of the above-mentioned studies used purified or synthetic model precursor proteins in an in vitro approach. As expected, such experiments indicated that arrested proteins sequester translocases available for other precursor proteins and thus strongly inhibit import. Few of the early studies used the yeast model to investigate the effects of in vivo expression of mitochondrial translocase blocking model proteins. These studies reported both growth impairment and mitochondrial import defects [186,189,191]. Importantly, native precursors also can stall during import, largely depending on their individual properties [9,149].

Only very recent studies uncovered molecular responses that resolve such import failure cases. A study in yeast proved that a pathway closely related to ER-associated protein degradation (ERAD) also operates on proteins that stall during mitochondrial import [196]. This newly discovered pathway, named mitochondrial protein translocation-associated degradation (mitoTAD), is constitutively active and restores the mitochondrial import in a UPS-dependent manner (Figure 5A). Similarly to ERAD, a ubiquitin-dependent molecular chaperone Cdc48/p97/VCP is recruited for extraction of the client protein. The pivotal component of the pathway is mitochondrially anchored Ubx2 protein that can bind to TOM translocon and recruit Cdc48 in complex with Npl4 and Ufd1 cofactors. Subsequently, AAA ATPase activity of Cdc48 powers the protein extraction from the translocation channel, allowing subsequent transfer to the proteasome for degradation [196,197,198,199].

The continuous action of mitoTAD maintains import in normal conditions. However, once the proteotoxic stress exceeds the capabilities of mitoTAD, different responses are activated, providing additional means to restore the import. One such mechanism that directly reacts to translocase overload with precursor proteins, termed mitochondrial compromised protein import response (mitoCPR), was described in yeast (Figure 5B) [9]. In this pathway, under prolonged clogging stress conditions, transcription factor Pdr3 induces the expression of Cis1 protein. Cis1 localizes close to the OM and is responsible for recruiting Msp1 AAA ATPase to clogged translocase. Finally, Msp1 extracts the stalled precursor protein, allowing its further degradation by the proteasome [9,200].

Both mitoTAD and mitoCPR were described in yeast. Ongoing research will clarify how responses to failed import events are orchestrated in other eukaryotic cells. Interestingly, key components of translocase unclogging were described in another context of mitoprotein quality control. Both Cdc48/p97/VCP and Msp1/ATAD1 contribute to the extraction of mislocalized or damaged proteins from the OM in a process called mitochondria associated degradation (MAD) (Figure 5C) [201,202,203]. In this mechanism corresponding to mitoTAD and mitoCPR, several E3 ubiquitin ligases, such as Mdm30 and Rsp5 in yeast or MARCH5/MITOL in mammals, are responsible for ubiquitination of OM substrates [204,205]. Marked proteins are released from the OM by Cdc48/VCP (yeast/mammals) with the help of Udf1, Doa1, or Vms1 proteins and driven for proteasomal degradation. Likewise, mislocalized tail-anchored proteins are extracted via Msp1 or ATAD1 in yeast and mammals, respectively. Here, proteins extracted from OM might be reinserted into the ER membrane, allowing their clearance through ER-associated degradation (ERAD) [203,206,207].

The machinery of MAD appears universal from yeast to human cells and is mechanistically similar to translocase unblocking machinery defined in yeast. Thus, an analogous mechanism is likely to safeguard translocases also in other eukaryotes. Besides that, inducible mechanisms such as mitoCPR operate in connection with previously described responses to precursor mislocalization. For instance, Rpn4, which is involved in mitoprotein-induced stress response by inducing the transcription factor Pdr3, may also activate mitoCPR [9,149].

The relevance of the mechanisms that unclog translocases extends beyond mitochondrial precursor proteins. Mitochondrial translocases can engage proteins that were not destined for the organelle and thus did not evolve to sustain translocation, posing an increased risk of clogging. Such events were reported in malignant tissues affected by neurodegenerative disorders such as Alzheimer’s and Parkinson’s diseases. In human-derived samples, variants of amyloid precursor protein (APP) were bound to mitochondria from the affected brain but not to mitochondria from the control brain [180,208]. APP was shown to interact with translocases of both the OM (TOMM40) and the IM (TIMM23) [180]. Furthermore, APP engages mitochondrial import machinery and stalls during translocation when overexpressed in cultured cells, resulting in general perturbation of mitochondrial import [209]. Finally, amyloid β peptides were also shown to interfere with the processing of mitochondrial proteins, resulting in the accumulation of precursor forms of such proteins [210].

Similarly, in Parkinson’s disease, mutant forms of α-synuclein, a small but very abundant soluble protein in the nervous system cells, were associated with mitochondrial membranes [211]. Furthermore, some species of α-synuclein interacted with TOM translocase component Tom20 and inhibited the import of mitochondrial precursor proteins [212]. In these clinical examples, non-mitochondrial proteins that are prone to aggregation were engaging the mitochondrial machinery increasing the clogging hazard.

Other recent findings might provide a twist in the interaction of mitochondrial translocases and aggregates. First, aggregates formed by newly synthesized proteins were observed tethered to mitochondria [183]. Next, aggregated proteins were shown to interact directly with the components of TOM translocase: Tom40 and Tom70 proteins [213]. Unexpectedly, some aggregated proteins were gradually imported and finally reached the mitochondrial matrix. Cytosolic chaperones appeared to regulate this process, with the Hsp104 disaggregase assisting mitochondrial uptake of aggregating proteins. By contrast, another cytosolic chaperone, Hsp70, seemed to counteract the import of aggregating proteins [213]. The mitochondrial import of protein deposits was proposed to serve as a clearance pathway [213].

In all of the above situations, mitochondrial import machinery is challenged with misfolding-prone or aggregated proteins. This stresses the need for efficient translocase unclogging that extends beyond natural precursor proteins and appears vital for preventing disease development.

## 9. Quality Control in Co-Translational Targeting

As discussed earlier, mitochondrial precursor targeting and import occur post-translationally or in a co-translational manner. In the second case, the precursor import that has already started cannot be completed until the synthesized protein is released from the ribosome. Thus, while the co-translational mechanism shields vulnerable precursors from the cytosolic environment, it creates a risk of translocon clogging due to translation stalling or failed translation termination. Such clogging will have matching consequences to these caused by folded precursor proteins. Due to the engagement of the ribosome, a dedicated ribosome-associated quality control system (RQC) acts to resolve such issues.

When translation stalls on the ribosomes in the cytosol, RQC removes nascent polypeptides in a ubiquitin- and proteasome-dependent manner. Initially, this requires 40S and 60S ribosome subunits to dissociate, giving access to the stalled polypeptide [214,215]. Subunit dissociation is mediated by ribosome rescue factors Dom34 (PELO), Hbs1 (HBS1L, GTPBP2), and Rli1 (ABCE1) [214,216,217,218]. The 60S subunit with stalled peptide is then recognized by ribosome quality control complex subunit 2 (Rqc2 in yeast, NEMF in human), which helps recruit E3 ubiquitin ligase Ltn1 (Listerin). In turn, Ltn1/Listerin ubiquitinate stalled substrates, providing the signal for extraction and degradation [219,220]. In addition to Ltn1 recruitment, Rqc2 runs a non-canonical synthesis reaction that extends the C-terminus of the stalled peptide by adding alanine and threonine residues—a process called CAT-tailing. Incorporating CAT-tails can further facilitate the ubiquitination of stalled substrates by Ltn1/Listerin [219,220]. The ubiquitinated substrates are removed from the 60S ribosome by Cdc48 and directed for proteasomal degradation. In yeast, CAT-tails were also shown to act as an additional degron sequence, recognized by Hul5 E3 ubiquitin ligase, further increasing the degradation efficiency [221].

The substrates that are co-translationally transported through membranes provide extra challenges for the RQC. Similarmechanism was first found for nascent ER peptides blocked in the Sec61 translocon [222]. This was followed by discovery of the mitochondrial import-specific RQC termed mitochondria-localized ribosome-associated quality control (mitoRQC) (Figure 2D). In the case of mitochondrial precursors, CAT-tailing has both positive and negative consequences. As a nascent precursor can be masked by mitochondrial translocase, the CAT-tail extension may help expose the ubiquitination site for Ltn1. However, in some cases, precursors with CAT-tail can escape degradation and complete import. Such proteins were shown to aggregate and disrupt proteostasis in the mitochondrial matrix [220]. To counteract such a pathological event, mitoRQC includes an additional mechanism that carries out early precursor degradation without CAT-tail addition. The peptidyl-tRNA hydrolase Vms1 (ANKZF1 in humans) competes with Rqc2 for the binding site and thus inhibits CAT-tailing, minimizing the pro-aggregative nature of this pathway [223,224]. Vms1 activity allows stalled precursors to be released. The released precursors can complete their import, becoming subject to intra-mitochondrial quality control, or can follow Ltn1-mediated degradation. Therefore, due to mitoRQC activity, stalled nascent precursors may be actively removed in a Cdc48- and UPS-dependent manner or directed into mitochondria, in both cases effectively unblocking mitochondrial translocases [225,226].

A recent report added another twist to co-translational quality control of mitochondrial precursors, with a phenomenon called mitochondrial-stress-induced translational termination impairment and protein carboxyl-terminal extension (MISTERMINATE) [227]. Authors show that mitochondrial dysfunction can impair the translation termination of nuclear-encoded mitochondrial precursors. In such a case, stalled ribosomes can continue beyond the stop codon extending the nascent peptide. This process is analogous to CAT-tailing, as added amino acid residues do not correspond to the mRNA sequence. A complex I protein, NDUFS3, was found to be prone to MISTERMINATE. The extended protein was imported into mitochondria, but it negatively affected respiratory function. Extended mitochondrial proteins were also shown to disturb cellular proteostasis. Ameliorating the RQC functions was shown to prevent MISTERMINATE and its negative consequences [227].

These data indicate that mitoRQC is another aspect of precursor protein quality control with a potentially high impact on understanding links between mitochondrial dysfunctions and protein folding disorders.

## 10. Conclusions and Perspectives

Traditionally, the process of mitochondrial proteome biogenesis is considered from the side of the organelle. This approach focuses on the organellar protein translocation pathways as well as internal quality control, folding, and maturation mechanisms. These are absolutely essential steps that ensure the proper distribution and function of mitochondrial proteins. However, the role of the cytosol cannot be reduced to a simple source of precursor proteins. From the moment of their synthesis, precursors are chaperoned, guided, and tightly controlled. Only the proper coordination of these external and organellar actions yields successful import.

Recent discoveries emphasized the impact of mitochondrial precursor proteins on the global proteostasis of the cell. Proteins that only transit the cytosolic environment can accumulate, mislocalize, or misfold. Without remedy, such incidents can quickly spread beyond mitochondrial precursors, creating a snowball effect. Growing evidence substantiates the vital role and mechanistic details of key cytosolic proteostasis pathways in the maintenance of precursor proteins.

Many of the described mechanisms rely on ubiquitination and proteasome-mediated degradation. However, there are other quality control pathways with a less clear impact on precursor proteins. Recent discoveries point to the generation of mitochondria-derived vesicles (MDV) or mitochondria-derived compartments (MDC) as complementary mechanisms to remove defective proteins from the organelle [228,229,230]. Such vesicles contain membrane proteins, including TOM components, and can be delivered to lysosomes for degradation. The formation of MDVs or MDCs could provide an alternative way to remove translocases with stalled precursors. This could be effective, as translocases are distributed in groups and thus could be removed in bulk [194]. However, further research is required to evaluate the role of vesicles in resolving failed precursor translocation.

When there is significant damage to mitochondria, entire organelles are degraded via mitophagy [231,232,233]. In malfunctioning mitochondria, where import is also defective, unimported precursors accumulate on the OM surface. By removing such organelles, mitophagy contributes to the precursor clearance. Furthermore, independently from mitochondria, autophagy can also degrade deposits of aggregated proteins [234]. Thus, if precursors aggregate in the cytosol, autophagy could clear these deposits. Future research will dissect the bulk degradation impact on precursors from the effect on mature mitochondrial proteins.

In the current review, we do not address mechanisms that maintain precursors inside the mitochondria. Mitochondria contain several members of the major chaperone families, which are directly implicated in the protein import or have essential functions in folding mitochondrial proteins. Gamitrinib, an inhibitor of the matrix Hsp90 TRAP1, was shown to strongly suppress mitochondrial biogenesis [235]. A starting clinical trial (NCT04827810) will test the potential of this drug in cancer therapy.

Mitochondria have multiple proteases that degrade precursors if their targeting or folding were not successful [236]. As described in earlier sections, mitochondrial proteases are also a part of regulatory circuits that respond to mitochondrial stresses. Interestingly, precursors directed to the IMS via the MIA pathway can be retro-translocated back to the cytosol if their oxidative folding is unsuccessful [237]. Such released precursors become cleared by the UPS, showing that the role of cytosolic machinery may extend to proteins that traversed the OM.

By focusing on precursors’ fates before they enter the organelle, we did not address the transcriptional regulation that precedes protein synthesis. The regulation of the levels of mRNAs encoding mitochondrial proteins influences their production volume. Transcripts encoding the most abundant mitochondrial proteins, including respiratory chain components, are downregulated in response to import failure [149]. Thus transcriptional regulation has to be recognized among contributors to mitochondrial proteome biogenesis.

Along with the understanding of how much mitochondrial precursor proteins can affect cellular proteostasis, their relationship with disease pathogenesis seems very likely. While current research indicates possible implications of precursors in proteostasis-related disorders, further studies should answer how direct and significant the association is with disease development.

Apart from global effects on proteostasis, cytosolic quality control was shown to impact specific proteins whose import is slower. Such situations may arise due to pathological mutation that affects precursor import rather than its function. Such proteins can be degraded too quickly and thus will not reach their destination [112,134]. Recent work proposed to repurpose proteasome-inhibiting drugs as a way to increase the import of such proteins into the mitochondria and thus recover their functions [134]. Possibly also other mutant mitochondrial proteins could be rescued by this approach [238]. Improving our understanding of the cytosolic quality control of mitochondrial biogenesis opens new possibilities to modulate organelle function.

## Figures and Tables

**Figure 1 ijms-23-00007-f001:**
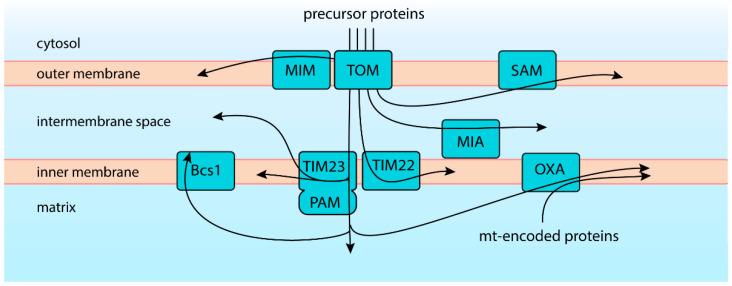
Precursor proteins follow various import routes. Proteins cross the outer membrane via TOM translocase. Outer membrane proteins are inserted straightway by MIM or through intermembrane space by SAM. Intermembrane space proteins are handled by MIA, which combines protein import and oxidative folding, or alternatively, diverge from TIM23 pathway. TIM22, TIM23, and OXA mediate the insertion of proteins into the inner membrane. Precursors destined for the matrix use TIM23 with PAM motor.

**Figure 2 ijms-23-00007-f002:**
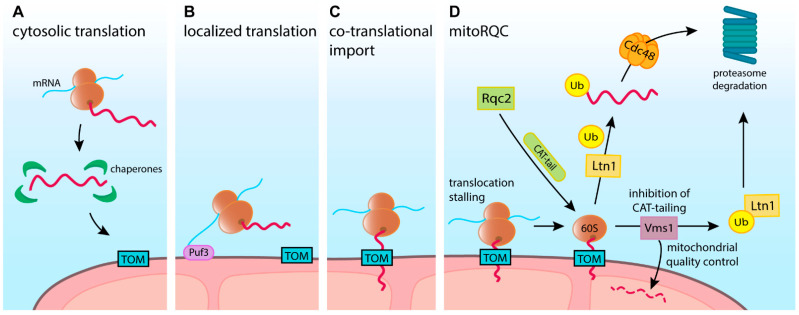
Location of mitochondrial precursor proteins’ synthesis affects their import mode. (**A**) Cytosolic translation and post-translational import expose precursors to the cytosolic environment. Precursors require guidance by chaperones, which prevent their misfolding or premature degradation. (**B**) Localized translation with ribosome or mRNA binding to the outer membrane by proteins such as Puf3, decreases the import distance. (**C**) Co-translational import protects precursors from the cytosolic environment. (**D**) MitoRQC responds to translation stalling during co-translational import. Rqc2 (ribosome quality control complex subunit in yeast) is adding a CAT-tail to the nascent precursor to facilitate its ubiquitination by Ltn1 ligase. Subsequently, ubiquitinated protein is guided by Cdc48 to proteasomal degradation. Alternatively, Vms1 has the ability to inhibit CAT-tailing and allows stalled precursors to alternate between proteasomal degradation or mitochondrial quality control.

**Figure 3 ijms-23-00007-f003:**
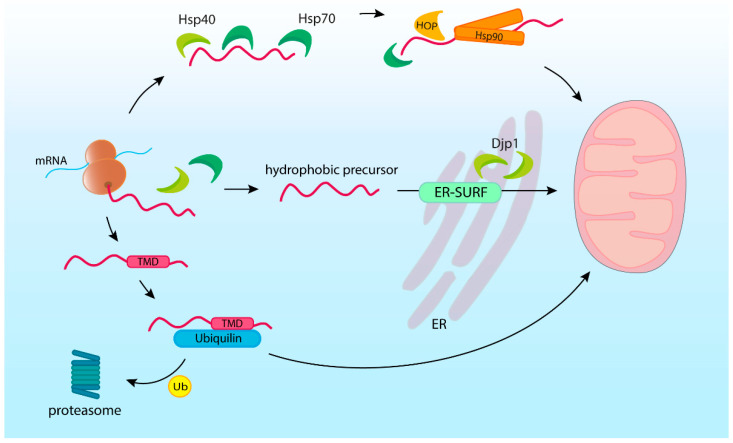
Molecular chaperones in post-translational import. Hsp40 and Hsp70 chaperones maintain mitochondrial protein precursors from the moment of their synthesis. At later stages Hsp90s, together with co-chaperone organizing proteins such as HOP, are also involved. Chaperones maintain precursors in an unfolded state and target them to mitochondria. ER surface-located chaperones such as Djp1 provide the environment that protects hydrophobic precursors in transit (ER-SURF). Ubiquilins bind trans-membrane domain (TMD) containing precursors and either allow their mitochondrial import or send them for degradation by the proteasome.

**Figure 4 ijms-23-00007-f004:**
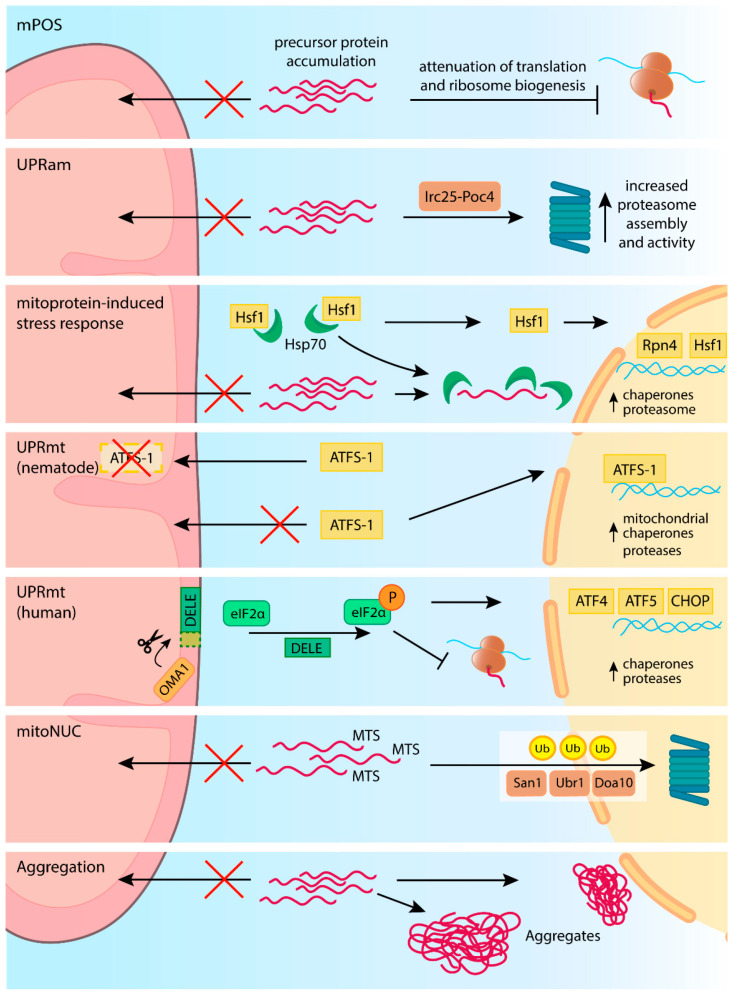
Mitochondrial import deficiency leads to proteotoxic stress. Defective import leads to the cytosolic accumulation of precursors, resulting in mitochondrial precursor overaccumulation stress (mPOS). Among the responses to mPOS, cells decrease translation and ribosome biogenesis to reduce the overload of proteins. Precursor accumulation also triggers unfolded protein response activated by mistargeting of proteins (UPRam) to increase proteasome assembly and activity. The mitoprotein-induced stress response is a transcriptional reprogramming that depends on heat shock transcription factor (Hsf1), which under stress conditions is released from chaperone complexes. Hsf1 will induce the expression of Hsp proteins, and through Rpn4, also proteasome levels. Mitoprotein-induced stress response also reduces levels of precursor-encoding mRNAs. Mitochondrial unfolded protein response (UPRmt) in nematodes reacts to import impairment in an ATFS-1-dependent manner. ATFS-1, which physiologically is imported and degraded in mitochondria, under the stress condition, localizes to the nucleus to increase the production of mitochondrial chaperones and proteases. UPRmt in humans depends on the proteolytic release of DELE1 protein by mitochondrial protease OMA1. In the cytosol, DELE1 promotes eIF2α phosphorylation, which modulates translation to trigger gene expression reprogramming. MTS-containing precursors that accumulate in the cytosol can be directed to the nucleus for proteasomal degradation in the nuclear-associated mitoprotein degradation (MitoNUC) pathway. Degradation is mediated by San1, Ubr1, and Doa10 E3 ubiquitin ligases. If not degraded, mislocalized precursors misfold and form aggregates.

**Figure 5 ijms-23-00007-f005:**
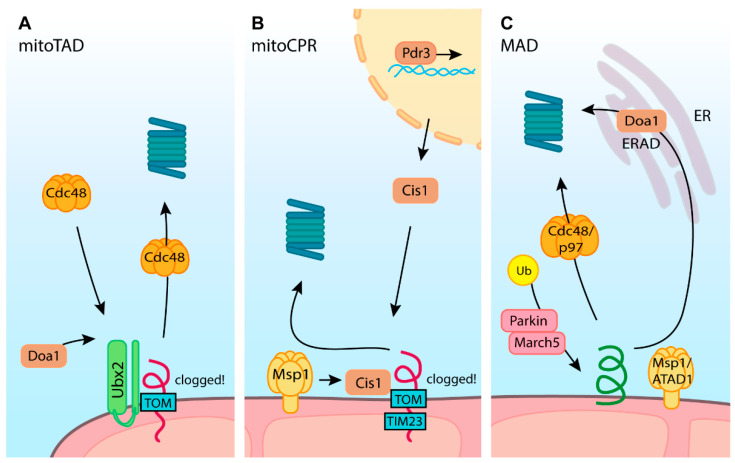
Quality control of blocked translocases. (**A**) In mitochondrial protein translocation-associated degradation (mitoTAD), Ubx2 with Doa1 recruits Cdc48 that transfers clogged precursor protein to proteasomal degradation. (**B**) In mitochondrial compromised protein import response (mitoCPR), import failure stress triggers the expression of Cis1 through transcription factor Pdr3. Cis1 localizes to the blocked translocases, anchoring the Msp1 AAA ATPase. Msp1 extracts blocked precursor proteins that the proteasome can degrade. (**C**) Mitochondria associated degradation (MAD) provides quality control to mislocalized or damaged outer membrane proteins by means analogous to translocase unblocking. Proteins are extracted either by Cdc48 (p97 in humans) or by Msp1 (ATAD1 in humans). Extracted proteins are processed by the ubiquitin-proteasome system.

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
