# Peer review of "Cytosolic Quality Control of Mitochondrial Protein Precursors—The Early Stages of the Organelle Biogenesis"

_ijms, 2021, doi:10.3390/ijms23010007_

Round 1
Reviewer 1 Report
Comments for the authors
The review article written by Anna M.Lenkiewicz et al., has described the cytosolic quality control of mitochondrial protein precursors. The article has provided the overview of cellular pathways that are involved in the precursor maintenance and guidance at the early stages of mitochondrial biogenesis. Overall, the manuscript is well written, and the figures are nicely drawn. Below are some of my comments for the authors.
- I think authors it is important to elaborate the role of mitochondrial stress signaling in different health and disease conditions.
- Also, it would be helpful to summarize in the table about the deregulation or mislocalization of mitochondrial proteins that are involved in the human diseases so far.
- How are mitochondrial precursor proteins routed between co- and post-translational import pathways in different organisms and cell types? Do ribosomes are encouraged or discouraged to synthetize mitochondrial proteins on the TOM complex?
Author Response
The review article written by Anna M.Lenkiewicz et al., has described the cytosolic quality control of mitochondrial protein precursors. The article has provided the overview of cellular pathways that are involved in the precursor maintenance and guidance at the early stages of mitochondrial biogenesis. Overall, the manuscript is well written, and the figures are nicely drawn. Below are some of my comments for the authors.
We thank the Reviewer for the thorough review, positive feedback, and comments that helped us improve the manuscript. Please find below our detailed responses to the comments:
- I think authors it is important to elaborate the role of mitochondrial stress signaling in different health and disease conditions.
Mitochondrial stress signaling impacts lifespan, health span, and pathology. This is a vast and important topic that, in our opinion, extends beyond the scope of our review. In our work, we focus on precursor proteins' fates from the moment of their synthesis until they enter the organelle. Thus we describe mostly the pathways that operate outside the organelle to maintain, guide, or remove precursors. Disturbed mitochondrial import can be both a cause and a result of mitochondrial stress.
To include more information on stress signaling, we now mention dedicated reviews that address this topic ((Bar‐Ziv et al., 2020; Tan & Finkel, 2020) (lines 534-537 with change tracking).
We have also added information on the implication of mitochondrial stress in protein aggregation-related conditions (lines 568-573).
- Also, it would be helpful to summarize in the table about the deregulation or mislocalization of mitochondrial proteins that are involved in the human diseases so far.
In the summary section of our work, we mention an example of the mutant protein, the precursor of which was shown to be degraded in the cytosol so quickly that its import became ineffective (Mohanraj et al.; lines 809-812). To provide more examples, we now also refer to a commentary by Habich and Riemer (2019), which lists other possible examples of such mutants. However, the exact relation between cytosolic quality control and mitochondrial import was not fully verified in these cases. Thus, we prefer not to include a table with the data that is still largely hypothetical and requires verification.
- How are mitochondrial precursor proteins routed between co- and post-translational import pathways in different organisms and cell types? Do ribosomes are encouraged or discouraged to synthetize mitochondrial proteins on the TOM complex?
We agree that it is an intriguing issue and highly significant for the topic. To date, no direct regulators that would sharply switch import modes between co-translational and post-translational were described in the case of mitochondria.
My personal experience with isolating mitochondria-associated ribosomes from yeast cells, indicated that this interaction is very transient. Treatment with cycloheximide, which arrests peptide synthesis, significantly extended this interaction (Gold et al. EMBO Rep. 2017). This observation suggests that interaction is based on nascent precursor protein. As the N-terminal mitochondrial targeting signal emerges from the ribosome exit tunnel, it can interact with import machinery. This is more likely for larger proteins. We describe this issue in lines 189 – 201.
As we mentioned in the manuscript, the co-translational import is more likely to occur when translated mRNA is localized close to the import site. Thus in the manuscript, we list currently known factors that aid localization of mRNAs to the mitochondrial surface in yeast, Drosophila fly, and human cells (lines 161-169). We have now added information about the study that suggests the existence of a factor that could have oposit effect by keeping mRNAs away from mitochondria (lines 197-198 - Vardi-Oknin D, Arava Y 2019).
With our current knowledge, we are unable to expand this part further. To highlight the fact of unclear regulatory mechanisms, we modified the fragment in lines 215-218 to read: "This is likely a regulated process that can vary depending on the cell type and its metabolic state. Most mitochondrial proteins can potentially switch between post- and co-translational import modes, but molecular determinants remain unclear."
Reviewer 2 Report
In this review the autors detail the different ways cells control and regulate mitochondrial protein precursors in the cytosol, describing the role of chaperones, different pathways involving the proteasome and getting into details as to how cells acheive a balance between precursors synthesised, imported, degraded and aggregating. Finally they discuss how these aspects may be important for the understanding of diseases. The review is very extensive, giving a lot of information both on what is known in yeast and other eucaryotes and bringing clear figures to come along with the text. The text is very well written in a logical order, figure legends are detailed and the discussion addresses the numerous questions which remain open and points still to be investigated.

Author Response
In this review the authors describe how the mitochondrial protein precursors, which are synthesised in and in part transported through the cytosol, are brought into the mitochondria. Thereby, they focus on the regulation at the cytoplasmic level, detailing different processes: achieving an equilibrium between translation of proteins and their translocation into the organelle, protection and control by chaperones at the ER/cytoplasmic/mitochondrial level and involvement of the proteasome which helps avoiding accumulation of precursors in the cytoplasm but also blocking of translocase of the outer/inner membranes by misfolded or misdirected proteins. In a second part, the authors get to the symmetric aspect of the question: how do mitochondrial precursors affect cytosolic proteostasis, what happens in stress or pathological conditions when mitoproteins accumulate in the cytoplasm – and they detail the cell’s options of regulation of protein turnover, shutting down of translation, specific synthesis of chaperones, proteasome subunits etc., increase of proteasome assembly... Finally, the authors address the consequences of mitochondrial protein precursor aggregation, which helps regulating the amount of proteins in the cytoplasm but also increases aggregation of proteins, some involved in diseases.
This review brings extensive information on the topic of cytosolic regulation of mitochondrial protein precursors and is very well written. The figures are clear, correlate nicely with the text and the legends always very complete. In the discussion, the authors summarize what has been done, the remaining open questions and ongoing investigations. This is also done sometimes within the different parts of the paper, which makes it even more interesting to the reader and helps putting the huge amount of information into perspective since there still are lots of unknowns.
Minor points.
l.128: “Such precursors are reach in hydrophobic”: “reach” should be “rich”
Fig4: “attenuation” misses one “t”
There is no title 9, it gets directly from 8 to 10.
We thank the Reviewer for the positive feedback and thorough review. We are grateful for finding typographical errors, which we have corrected in the revised manuscript according to the Reviewer's suggestion.